# VISUALIZING REPRESENTATIONS BY PERCEPTUAL REGULARIZATION

## ABSTRACT

A deployable machine learning model relies on a good representation. A desirable criteria for a good representation is to be interpretable by humans. We propose a technique termed perceptual regularization that enables visualization of the latent representation. By visualizing the learned representation, we are also able to understand the attention of a model, obtaining visual evidence that supervised networks learn task-specific representations. Moreover our method provides a direct visualization of the effect that adversarial attacks have on the internal representation of a deep network. We show how this can be used to systematically perform latent space interpolation to modify semantic features.

## 1 INTRODUCTION

Deep neural networks are known to be vulnerable, small changes known as adversarial attacks can completely change the model's prediction (Szegedy et al., 2014; Nguyen et al., 2015; Kurakin et al., 2017). There is a rapidly growing body of work on how to perform effective adversarial attacks, and how to defend against them (Papernot et al., 2017; Madry et al., 2017; Athalye et al., 2018). On the one hand, adversarial examples are named as such because they as able to trick a deep network using changes imperceptible to humans. On the other hand, the model makes predictions based on its internal representation (i.e. what the model perceives) which may be very different from human perception. Indeed, the existence of adversarial examples is evidence that human perception and machine perception are misaligned, and the internal representation of a model and a human differ significantly. The motivating question we would like to address is: are adversarial examples really adversarial examples from the model's perspective?

In order to get insight of this question, we extend an existing visualization technique from Dosovitskiy & Brox (2016b); Johnson et al. (2016); Dosovitskiy & Brox (2016a), stacking a decoder head on top of the model's latent representation with the aim of providing a human interpretable reconstruction, see Figure 1. In particular, we train the decoder jointly with the network, terming the resultant objective function *perceptual regularization*. A similar strategy has also been explored in Sabour et al. (2017); Qin et al. (2019) for Capsule networks, and exploiting semi-supervised (unlabeled) data in Lasserre et al. (2006); Grabner et al. (2007); Larochelle & Bengio (2008); Le et al. (2018). One of the key insights we bring in our work is that a simple decoder head term is all that is needed to yield significant visual insight into a range of questions about the representations learned by deep networks.

Visualization is an active research area, with various standard techniques such as saliency maps Simonyan et al. (2014) among others Zeiler & Fergus (2014); Mahendran & Vedaldi (2015); Springenberg et al. (2015); Zhou et al. (2016) being used to visualize representations by inverting them. However, reconstructing the original input from the representation is computationally expensive. An alternative way is to introduce a forward decoder from latent space to image space, aiming to globally invert the representation. This technique has been used in Dosovitskiy & Brox (2016b); Johnson et al. (2016); Dosovitskiy & Brox (2016a) and our method follows this line of work.

The main contributions of this paper are:

- Extending the existing visualization method as a regularization method for jointly learning a visualization for deep representations.

- Using this method to visualize how the attention of a model focuses only on aspects that are relevant for prediction. More generally we give promising evidence to suggest that our visualization method is of interest for model diagnostics and interpretability.

- Using this visualization method to shed light on what effect adversarial attacks have on a model.

- Using adversarial attacks to identify direction vectors in the latent space that correspond to certain semantic features. We show that the decoder network can then be used to synthesize images with modified semantic features.

## 2    PERCEPTUAL REGULARIZATION

We begin by introducing perceptual regularization. Consider a deep image classification model such as AlexNet (Krizhevsky et al., 2012), VGG-Net (Simonyan & Zisserman, 2015), or ResNet (He et al., 2016). It usually consists of two stages: several convolutional layers, followed by dense layers. The convolution layers can be viewed as a feature extraction process which we denote by $\mathcal{F}$, transforming images from the input space $\mathcal{X}$ into a latent space $\mathcal{Z}$. The dense layers $\mathcal{C} : \mathcal{Z} \to \mathcal{Y}$ act as a classifier on the latent space. In order to regularize and visualize the latent representation, we introduce a decoder $\mathcal{D} : \mathcal{Z} \to \mathcal{X}$ on top of the latent space $\mathcal{Z}$ which we jointly train with the classification model. The architecture is shown in Figure 1.

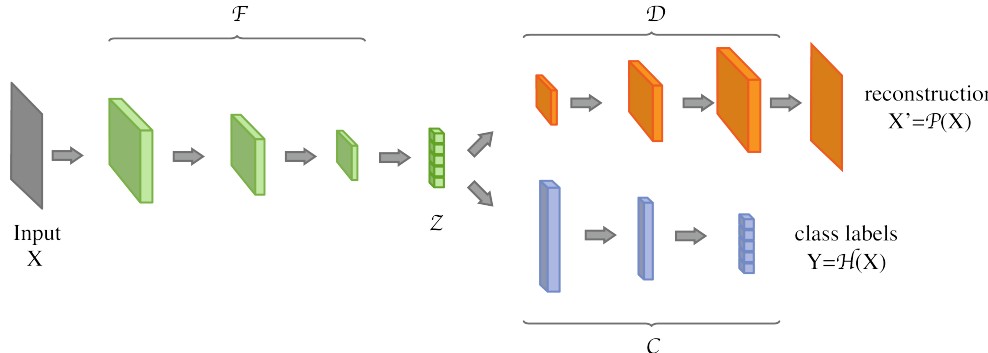

Figure 1: Perceptual regularization: we stack a decoder on top of the feature map and jointly train it with the classifier.

More precisely, we denote the classification model by $\mathcal{H} = \mathcal{C} \circ \mathcal{F}$ and term the reconstruction $\mathcal{P} = \mathcal{D} \circ \mathcal{F}$ a *perception* of the model. The regularization strategy we introduce is to jointly train $\mathcal{H}$ and $\mathcal{P}$ together by minimizing

$$\min_{\mathcal{P},\mathcal{H}} \mathbb{E}_{X,Y} \, [\, \underbrace{\ell(\mathcal{H}(X), Y)}_{\text{classification loss}} + \lambda \underbrace{\|X - \mathcal{P}(X)\|^2}_{\text{reconstruction loss}}]. \qquad \text{(Perceptual regularization)}$$

The first term is the standard classification loss and we take $\ell$ to be the cross-entropy loss throughout the paper. The regularization term is the $\ell_2$ reconstruction loss scaled by a hyper-parameter $\lambda > 0$. Intuitively, the decoder tries to reconstruct the original image based on the same information the classifier uses for prediction. In particular, if the feature map $\mathcal{F}$ preserves all the information from the input, the perception should be able to perfectly recover $X$. Conversely, if a lot of information has been thrown away, then we will get a poor reconstruction and the regularization loss will be high. Therefore, our regularization biases the learned representation to maintain more information about the input compared to vanilla supervised learning. This will be useful if one would like to reuse the learned features later for other tasks.

Notice that the regularization term vanishes when $\mathcal{P}$ is the identity mapping, in which case the feature map $\mathcal{F}$ is bijective. This can happen when the model is very powerful and therefore able to memorize any dataset. However, this situation is less interesting in the sense that a good representation should efficiently compress the information. This idea has been clearly understood with the information bottleneck objective, for which it is known that $\mathcal{F}(X)$ approximates the minimal sufficient statistic for $Y$ given $X$ (Shamir et al., 2010). Therefore, to constrain the expressiveness of the feature map, we impose a bottleneck structure in the network architecture. More precisely, the dimension of the

latent space $\mathcal{Z}$ is made significantly smaller than the dimension of the input space $\mathcal{X}$. This enforces the feature map to compress information and makes it very unlikely to be bijective.

As well as regularizing the model, perceptual regularization also provides a natural way to visualize the latent space. In particular, the perception $\mathcal{P}$ is designed to decode the feature map $\mathcal{F}$, which is very useful for model understanding. In the next three sections we begin to explore the insights that can be drawn using perceptual regularization.

## 3 VISUALIZING FEATURES LEARNED BY A DEEP NETWORK

Extensive research have been conducted on attention-based models in object recognition and image captioning Bahdanau et al. (2014); Karpathy & Fei-Fei (2015); Ba et al. (2015); Gregor et al. (2015); Xu et al. (2015); Jaderberg et al. (2015); He et al. (2017), where the attention mechanism is explicitly introduced as a spatial map highlighting image regions. A second application of perceptual regularization is to visualize what a classifier's attention focuses on. In particular, if a model is trained to predict the object in the middle of an image, it is commonly believed that the learned representation "overfits" to the training task by focusing only on the central part of the image. This can be a cause of negative transfer - when reusing a learned representation for a new task actually *hurts* performance (Pan & Yang, 2009; Kuzborskij & Orabona, 2013). We show our visualization method can help understand when a representation will cause negative transfer. All experimental details, including architecture and hyper-parameters are given in full in the appendix.

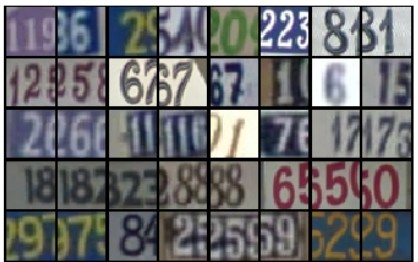 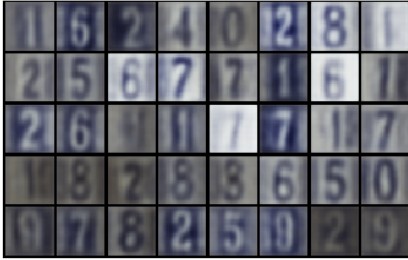

Figure 2: Left: Original images from SVHN. Right: The model's perception of the images.

### 3.1 THE ATTENTION PROBLEM

It has long been intuitively understood that the representations in the final layers of a supervised deep network will only focus on aspects of the data relevant to its prediction task (Long et al., 2015; Yosinski et al., 2014; Alain & Bengio, 2017). In this section we demonstrate on the SVHN dataset (Netzer et al., 2011) that this phenomenon of focused attention can be visualized using perceptual regularization.

The SVHN dataset consists of images of house numbers and the task is to predict the number in the center of the image. In particular, an image could contain multiple numbers but the label is given according to the middle one. A natural question is: will a supervised model learn a representation that remembers numbers that are not in the center? To answer this question, we train a CNN with perceptual regularization and then use perception to visualize the attention of the model, i.e. given any input $X$, we visualize $\mathcal{P}(X)$. As shown in Figure 2, our visualization technique shows that the model learns a representation that focuses on the middle of the image, and forgets what is at the edges of the image. This provides additional evidence to support the observation that a supervised model learns a representation specific to its prediction task. This is fine if there is only one task of interest, but makes transferring features between tasks problematic. For example, in our case, the representation would be completely ineffective for identifying the digit on the left or right hand sides.

### 3.2 PERCEPTUAL REGULARIZATION TO CONTROL ATTENTION

The cause of this restricted attention problem is the guidance given by the labels. In this section, we demonstrate that perceptual regularization can be useful to reduce the problem of overfitting learned features to the training task. In particular, our method allows for a smooth interpolation between supervised learning and unsupervised learning. In the extreme case, when $\lambda \to 0$, we recover

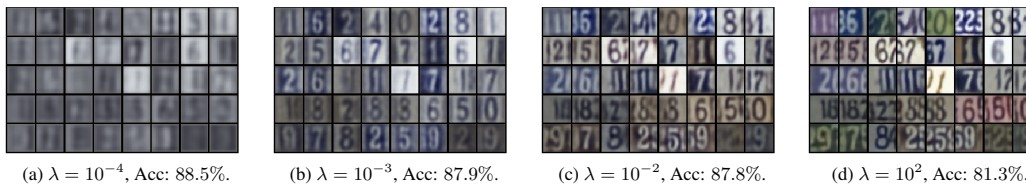

(a) $\lambda = 10^{-4}$, Acc: 88.5%.    (b) $\lambda = 10^{-3}$, Acc: 87.9%.    (c) $\lambda = 10^{-2}$, Acc: 87.8%.    (d) $\lambda = 10^{2}$, Acc: 81.3%.

Figure 3: Visualizing the model's attention. Acc is the accuracy of the classifier on the test set.

the standard supervised learning setting, and when $\lambda \to \infty$ we recover the classical auto-encoder objective. Hence, the parameter $\lambda$ can control how much information we would like our model's representation to keep.

We run experiments on the SVHN dataset by varying the regularization parameter $\lambda$. We keep the architecture fixed (see Appendix B.3 for more details). Figure 3 shows different regimes of features for different values of $\lambda$. There are four main regimes:

- Tiny ($\lambda < 10^{-4}$, Figure 3(a)): Since $\lambda$ is extremely small the contribution to the loss from the decoder branch is so negligible that there is not much incentive to learn a good decoder and the visualization breaks down.
- Delicate ($10^{-4} < \lambda < 10^{-2}$, Figure 3(b)): A good decoder is learned and the bias injected into the learned feature map by the regularization term is small. The number in the center is well reconstructed, but the edges of the image are not and the colors are missing. It is in this regime that we are able to see the restricted attention of a classifier.
- Intermediate ($10^{-2} < \lambda < 1$, Figure 3(c)): We obtain more faithful reconstructions compared to the "delicate" regime. In particular, the numbers at the edges are reasonably reconstructed but colors are still not well captured.
- Large ($\lambda > 1$, Figure 3(d)): The decoder is very good, both numbers and colors are recovered. The regularization has had a significant effect on the nature of the learned features. Since these features retain much more information than for smaller $\lambda$ they are more likely to transfer well to new unseen tasks.

Clearly the choice of $\lambda$ heavily affects the features learned by the model. The larger $\lambda$ is, the more information is preserved in the features: small values of $\lambda$ only preserve the number in the middle of the image, intermediate values can also reconstruct the boundary of the image, and only for large $\lambda$ is the input's precise color correctly reconstructed. It is also important to note that the accuracy on the supervised task decreases as $\lambda$ increases, but at a relatively slow rate. This is to be expected. In the limit $\lambda \to \infty$ there is no longer any incentive to perform good classification, only reconstruction. An interesting future direction will be to understand why certain features are harder to learn than others, for instance, in our example why the color is recovered later than numbers along the edges.

It is important to note that perceptual regularization introduces an inductive bias towards learning representations that retain a lot of information about the original input, which are different from what is learned without it. If one wanted to avoid this bias then one idea is to learn the feature map first, then learn the visualization decoder separately, with the feature map frozen. This strategy works reasonably well when the network's size is small. But, as the network gets more complicated it becomes computationally difficult to decode the feature map. Hence, as an alternative solution, we train the decoder jointly with the feature map. What we have shown is that although the bias induced by the regularization is an apparent drawback, it can in fact be turned into a strength by using $\lambda$ to control the behavior of the feature map.

Finally, the specific values of $\lambda$ quoted are meant only to serve as a very rough guide about trends. Making the magnitude of the reconstruction loss and classification loss comparable is often effective to locate the "delicate" $\lambda$.

## 4   HOW DO ADVERSARIAL ATTACKS AFFECT A CLASSIFIER'S PERCEPTION?

Having illustrated how the decoder network can be used to study the information content of a learned representation, we now turn our attention to adversarial examples. Adversarial examples can seem quite mysterious, and point to a way in which human and machine perceptions are misaligned (Han et al., 2019; Engstrom et al., 2019). While intuitively reasonable, understanding of this phenomenon

has been limited by the difficulty of observing the effect of adversarial attacks on the internal representation of a model (Zhang & Zhu, 2018; Olah et al., 2018). In this section we explore the first application of our perceptual regularization: to provide human understandable visualizations of a model's internal representation of adversarial examples.

First, we briefly recap how adversarial examples are obtained. Given a classification model $\mathcal{H}$ and a sample $(X, Y)$, we aim to fool the model's prediction by slightly perturbing the input $X$. We constrain the perturbation to be in a small neighborhood $\Delta(X)$ of $X$, which is usually defined as an $\ell_p$ ball. Specifically, we look for

$$X_{\text{adv}} \in \underset{X' \in \Delta(X)}{\arg\max} \ \ell(\mathcal{H}(X'), Y) \tag{1}$$

While many different strategies have been proposed to adversarially attack the model (Goodfellow et al., 2015; Moosavi-Dezfooli et al., 2016; Kurakin et al., 2017; Athalye et al., 2018), we focus on the projected gradient descent (PGD) attack (Madry et al., 2017) throughout this paper. Specifically, we set $\Delta(X)$ to be an $\ell_\infty$ ball and the adversarial example $X_{\text{adv}}$ is obtained by performing 100 iterations of PGD attack. An example of an adversarial attacks can be found in the first row of Figures 4. All experimental details, including architecture and hyper-parameters are given in full in the appendix.

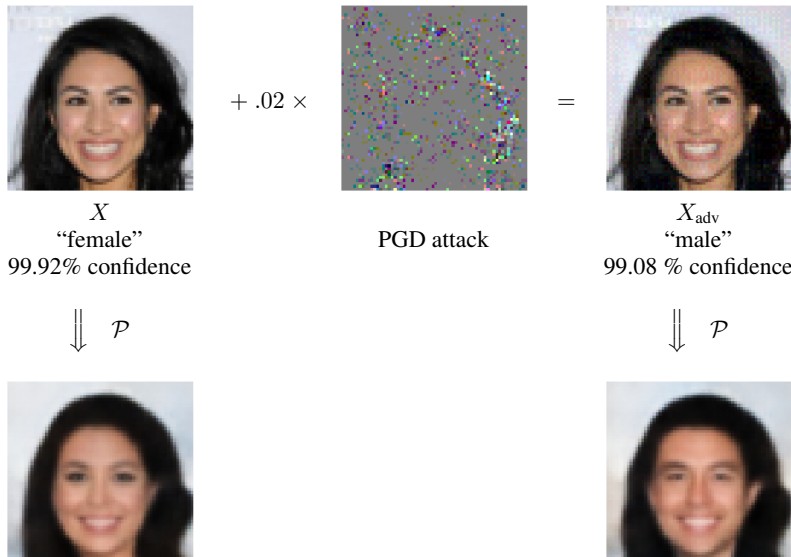

Figure 4: First row: by adding an imperceptibly small vector whose elements are obtained from performing a PGD attack, we can change the classification of the image. Second row: by applying perception, we can understand the model's "incorrect" prediction.

By design, adversarial examples are hard to distinguish from the original image with human perception. The perceptual regularization architecture gives a classifier, and a separate branch that reconstructs the original image. We adversarially attack the classification branch $\mathcal{H} = \mathcal{C} \circ \mathcal{F}$ using (1) to obtain $X_{\text{adv}}$. If, instead of viewing $X$ and $X_{\text{adv}}$, we look at the model's perception of each image $\mathcal{P}(X)$ and $\mathcal{P}(X_{\text{adv}})$, a clear disparity emerges between how the model understands the two examples. What is extremely striking is that the model's perception and incorrect classification are semantically consistent to the human eye. Put simply: *when the model misclassifies an image of a female as male, the perception of the adversarial image really looks like a male to a human*, as shown in Figure 4; *when the model misclassifies an image of a smiling face as not smiling, the perception of the adversarial image really looks like it is not smiling*, as shown in Figure 5.

We provide further experiments of our method on the MNIST (LeCun et al., 1998) and CelebA (Liu et al., 2015) datasets. Our perception decoder allows one to visualize the effect of an adversarial attack on a deep network's representation. For CelebA we train three different classifiers for three different binary classification tasks (Eyeglasses/Smiling/Gender). All experiments support the same observations made in this section. See the appendix for many more examples.

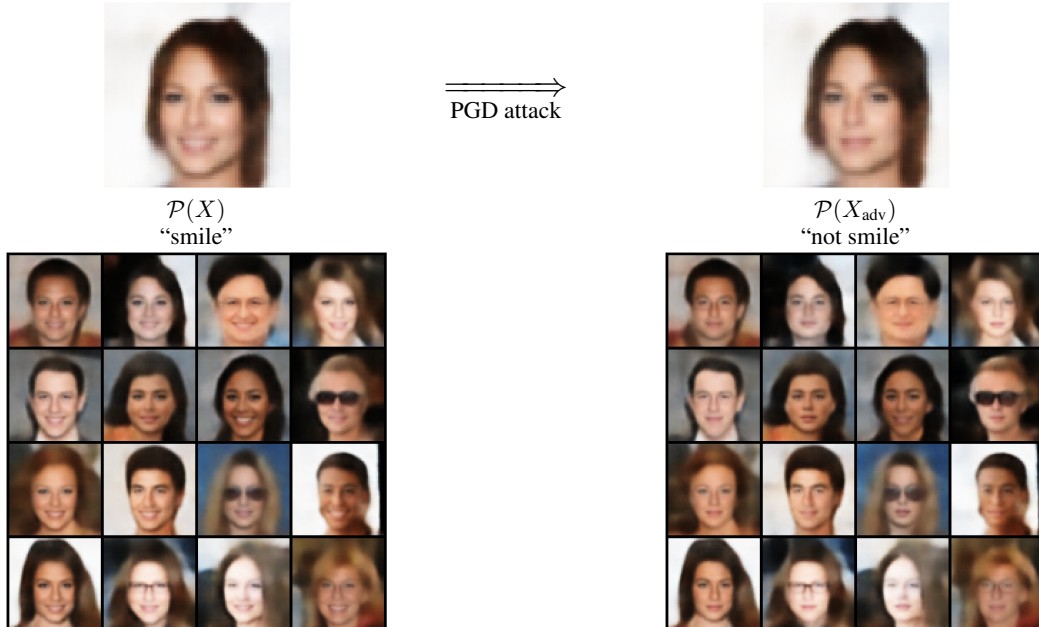

Figure 5: Images after applying perception on the task of classifying smiling face, we can understand the model's "incorrect" prediction. The corresponding original images can be found in Appendix A.

This shows that given the model's feature representation, it is making a prediction that appears reasonable on adversarial images. In other words, the representation learned by the model is very different from human perception. It is also interesting that the images obtained from perception (Figure 5, in the second row of Figure 4, ) are simultaneously close to each other and yet highly distinguishable in the characteristics most important for prediction. This gives interesting visual insight into the way in which adversarial attacks find a short path to a decision boundary. For example, Figure 4 has several subtle modifications: lighter eyes, squarer forehead and jaw, thicker nose. The changes are very targeted, capturing the essence of the global feature.

This begs the question: can adversarial attacks be used to identify and modify high-level features? Global features are high level human-interpretable features such as, "wearing glasses", "is smiling", or "is female"). There has been much previous work on learning disentangled representations for which the latent variables correspond to disentangled human-interpretable features (Tenenbaum & Freeman, 2000; Chen et al., 2016; Higgins et al., 2017). To then modify a given global feature one needs to figure out which latent feature (or which combination) to adjust. However, the task of finding the "correct" latent direction is non-trivial.

## 5 ADVERSARIAL ATTACKS FOR LATENT SPACE INTERPOLATION

Our visualization method yields a novel way for finding directions in latent space corresponding to a given global feature. Traditional approach applying generative adversarial network such as Radford et al. (2015); Karras et al. (2019) involve significant manual effort to identify which latent direction corresponds to a given semantic feature. Our method provide a more straightforward way to obtain this direction by performing adversarial attacks. The recipe is as follows:

- Train a model with perceptual regularization,

- Given $X$, obtain $X_{adv}$ by performing an adversarial attack toward the target label (e.g. smiling/wear eyeglasses),

- The latent space direction corresponding to this global feature is $Z = \mathcal{F}(X_{adv}) - \mathcal{F}(X)$,

- Synthesize images with adjusted global feature $\mathcal{C}(\mathcal{F}(X) + \alpha Z)$ for some $\alpha$.

Larger values of $\alpha$ will make larger adjustments to the global feature. Figures 4 and 5 give simple examples of this for $\alpha = 1$. Figure 6 shows images synthesized by following a linear path through latent space for a range of values of $\alpha \in [0, 2]$. The results show that our technique allows precise modification of the target global feature, leaving other aspects of the image unaltered.

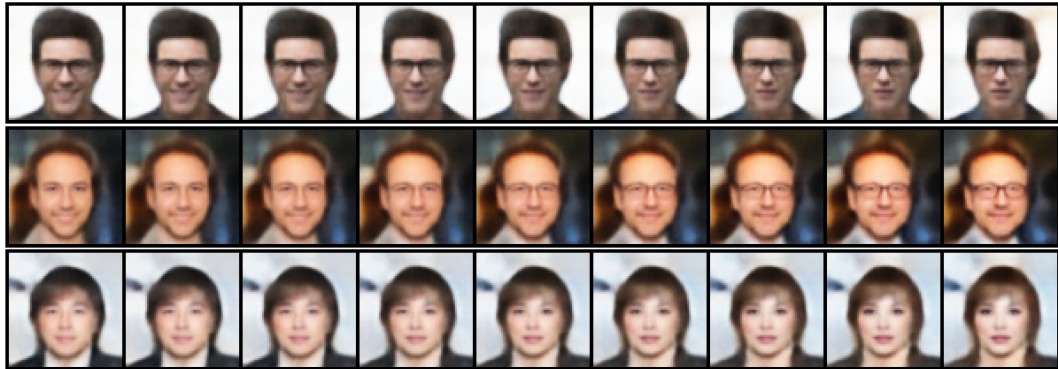

Figure 6: First row: smiling to not smiling. Second row: no glasses to glasses. Third row: female to male. Each column corresponds sequentially to $\alpha \in \{0, 0.25, 0.5, 0.75, 1, 1.25, 1.5, 1.75, 2\}$.

We believe it is possible to obtain more realistic images using recent progress on GANs, but we leave this for future work. Finally, we stress that our method requires labeled data while GANs do not. However, in situations where one has labeled data, our method provides a promising alternative since it does not require human intervention.

## 6 DISCUSSION AND CONCLUDING REMARKS

In this paper we introduced *perceptual regularization*, a method for visualizing representations and show that it can be used for a variety of purposes including: assessing the informational content of a representation, visualizing the effect of adversarial attacks on the latent representation, and identifying semantically meaningful latent space direction vectors. To achieve this, our method combines a generative and discriminative model. There has been previous work on the idea of combining generative and discriminative models (Lasserre et al., 2006; Grabner et al., 2007; Larochelle & Bengio, 2008; Le et al., 2018), however these works have focused on the idea of using such a model to exploit semi-supervised (unlabeled) data. To the best of our knowledge our work is the first to identify the generative plus discriminative formulation as an effective method for practical, visual appraisal of the representation learned by a model. We use this visualization to understand adversarial examples and study the attention of a model. We also identify this formulation as suitable for learning representations that are effective on previously unseen tasks.

In this work we made the choice to consider the $\ell_2$ loss as a metric between images. However, the specific nature of the injected bias is particular to $\ell_2$: it permits lots of small errors, but penalizes large differences severely. It would be an interesting future direction to consider different metrics and consider what kind of representations are learned by regularizing with different generative model objectives such as GANs, VAEs, or InfoMAX (Goodfellow et al., 2014; Kingma & Welling, 2014; Hjelm et al., 2019).

Another interesting question is to understand what conditions are necessary to obtain good visualizations. We identify two possible factors. First: model complexity. As the task becomes more challenging we may need a more powerful decoder and refined training strategy (e.g. applying methods from the GAN literature). Second: human meaningful features. There has been work suggesting that more complicated datasets have small but highly predictive features that humans do not notice (Tsipras et al., 2018). The model may learn to use these features for prediction. In this case our visualizations may also not be human meaningful. Nevertheless, our visualization method could be a useful certificate for whether the model learns human meaningful features.

This raises the question of how to learn robust features that generalize across multiple tasks. There is evidence suggesting adversarial training may help learn more "human-aligned" features. But so far work on adversarial training has mainly focused on supervised learning. We suspect that this will not

be sufficient to learn transferrable features. It would be an interesting next step to combine perception regularization with adversarial training.

Our visualizations raise a broader point about the supervised learning paradigm: models insist on pigeon-holing every input into one of a fixed number of classes, no matter how out of sample the input is. Consider the following toy example: taking $X$ to be random noise and computing the adversarial attack $X_{adv}$ for a model $\mathcal{H}$ trained on MNIST, then $\mathcal{P}(X_{adv})$ looks like a digit to the human eye (see Figure 7). This is undesirable and it is not sufficient to simply add an 11th "other" class. This is a long way from the dynamic way that humans create and learn new classes of objects.

To conclude, we have shown that perceptual regularization is a promising approach to answering many important questions in machine learning. However we believe this is just the beginning, and view perceptual regularization as a broadly useful tool for model diagnostics with many more uses to be uncovered in the future.

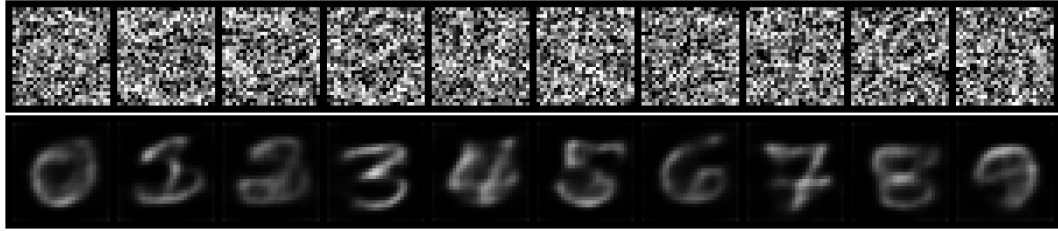

Figure 7: First row: $X_{adv}$ for random noise $X$. Second row: $\mathcal{P}(X_{adv})$. Even for what is essentially random noise, the model insists on perceiving something that semantically looks like a number.

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

## A   ADDITIONAL IMAGES ON ADVERSARIAL EXAMPLES

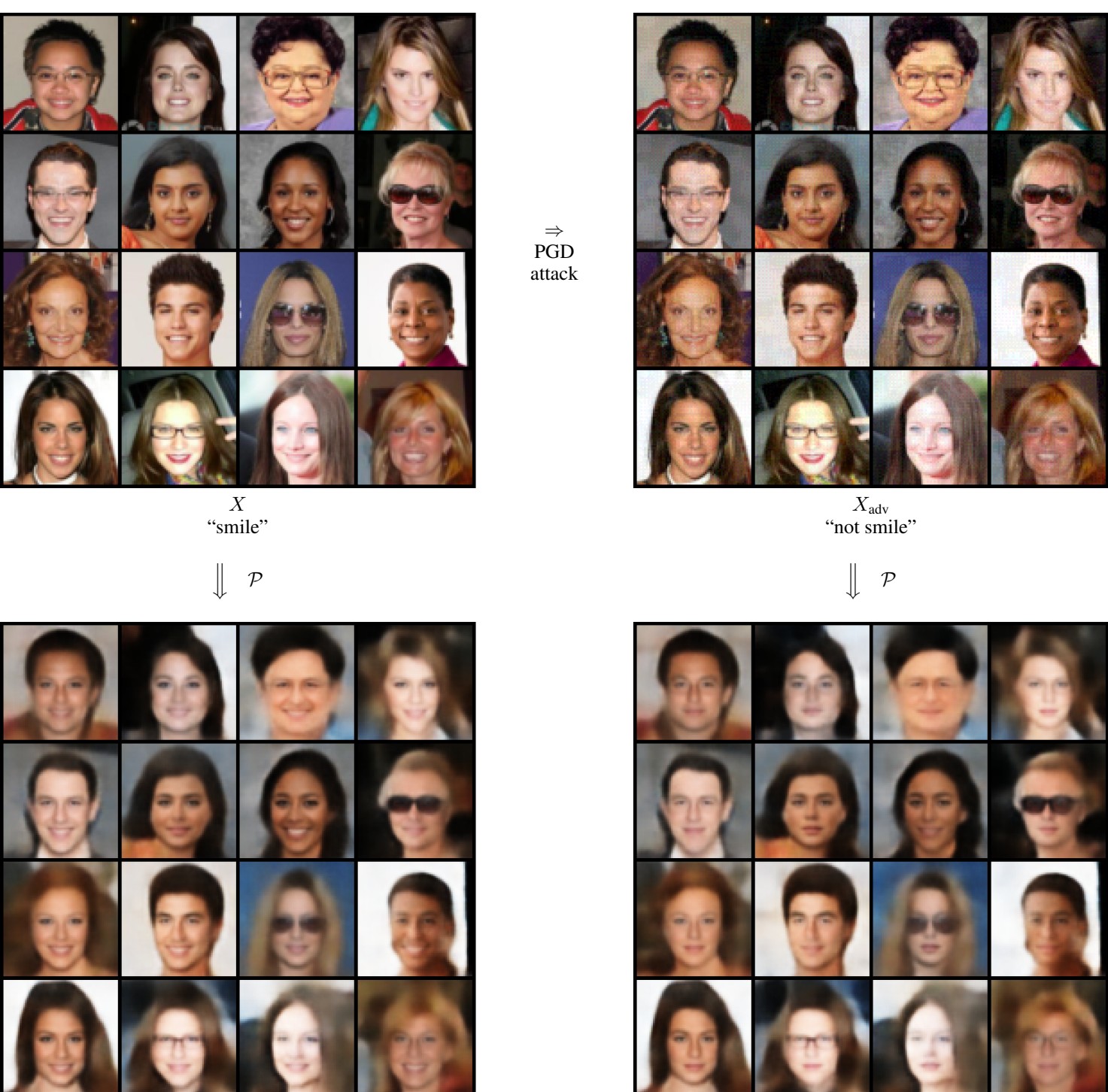

Figure 8: Visualization of adversarial examples on CelebA dataset with Smile labels. The adversarial examples are obtained by applying 100 iterations of PGD attack with $\ell_\infty$ perturbation and $\epsilon = 0.03$. The regularization parameter is set to $\lambda = 10^{-3}$.

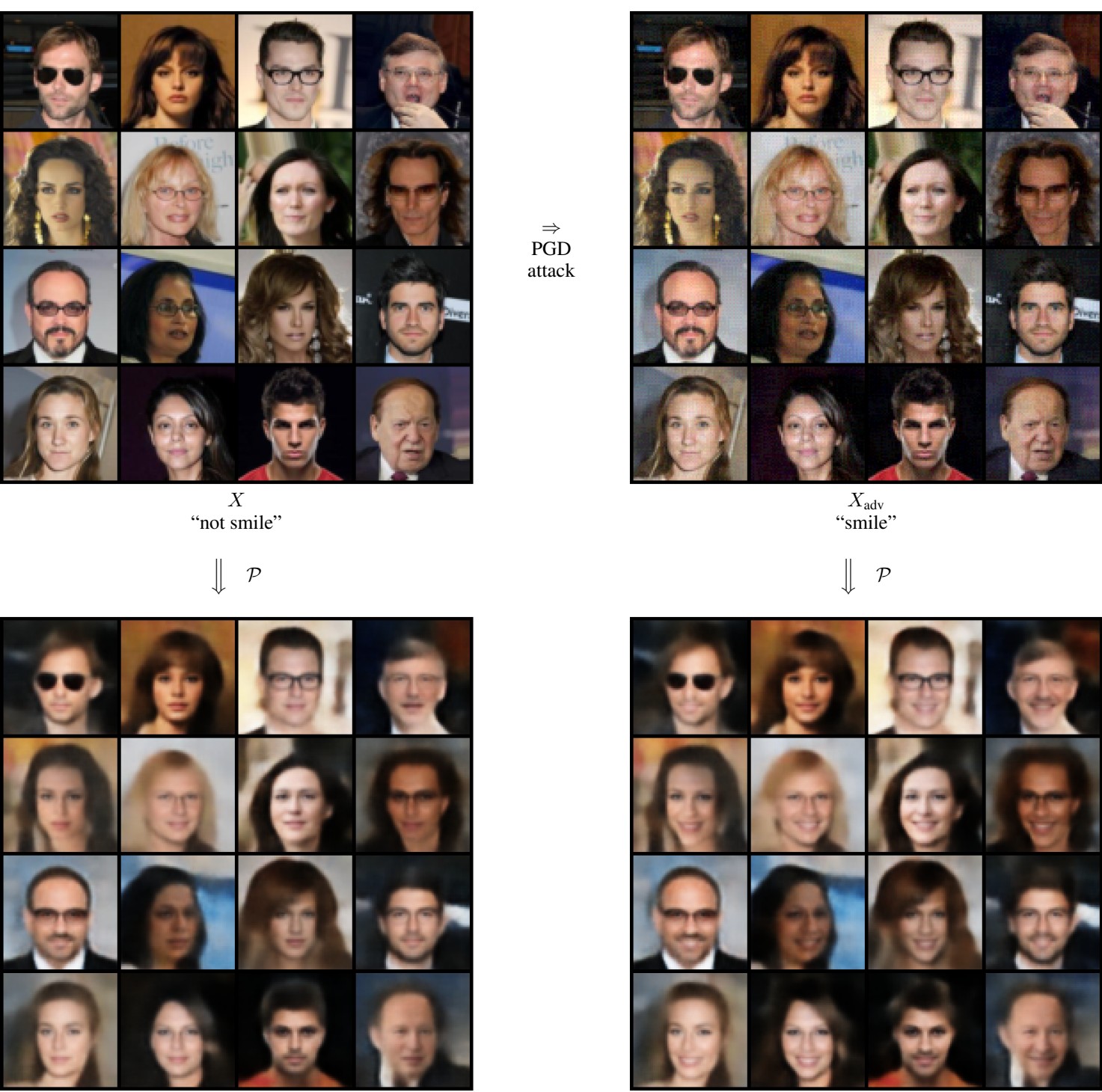

Figure 9: Visualization of adversarial examples on CelebA dataset with Smile labels. The adversarial examples are obtained by applying 100 iterations of PGD attack with $\ell_\infty$ perturbation and $\epsilon = 0.03$. The regularization parameter is set to $\lambda = 10^{-3}$.

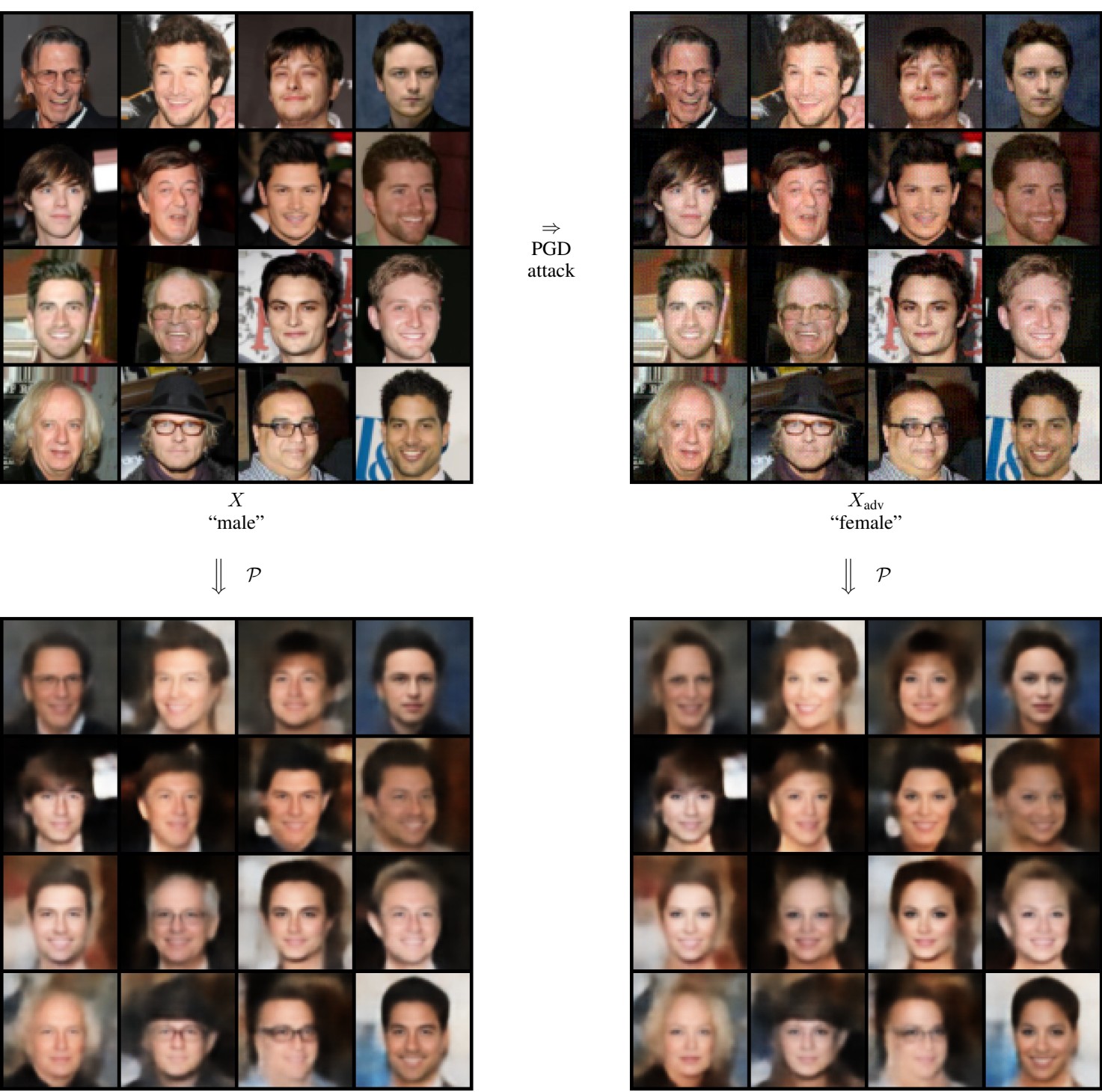

Figure 10: Visualization of adversarial examples on CelebA dataset with eyeglasses labels. The adversarial examples are obtained by applying 100 iterations of PGD attack with $\ell_\infty$ perturbation and $\epsilon = 0.03$. The regularization parameter is set to $\lambda = 10^{-3}$.

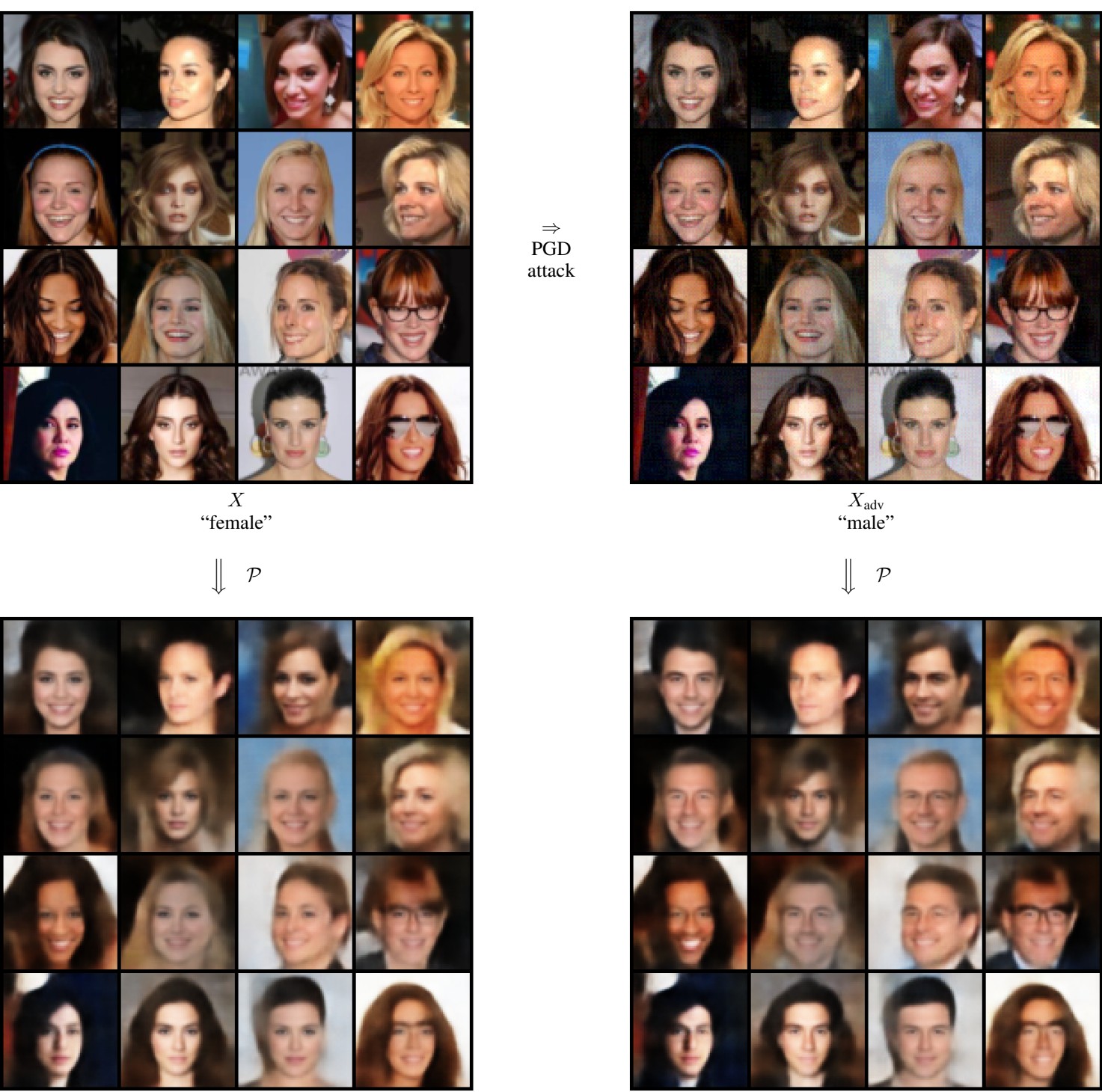

Figure 11: Visualization of adversarial examples on CelebA dataset with Gender labels. The adversarial examples are obtained by applying 100 iterations of PGD attack with $\ell_\infty$ perturbation and $\epsilon = 0.03$. The regularization parameter is set to $\lambda = 10^{-3}$.

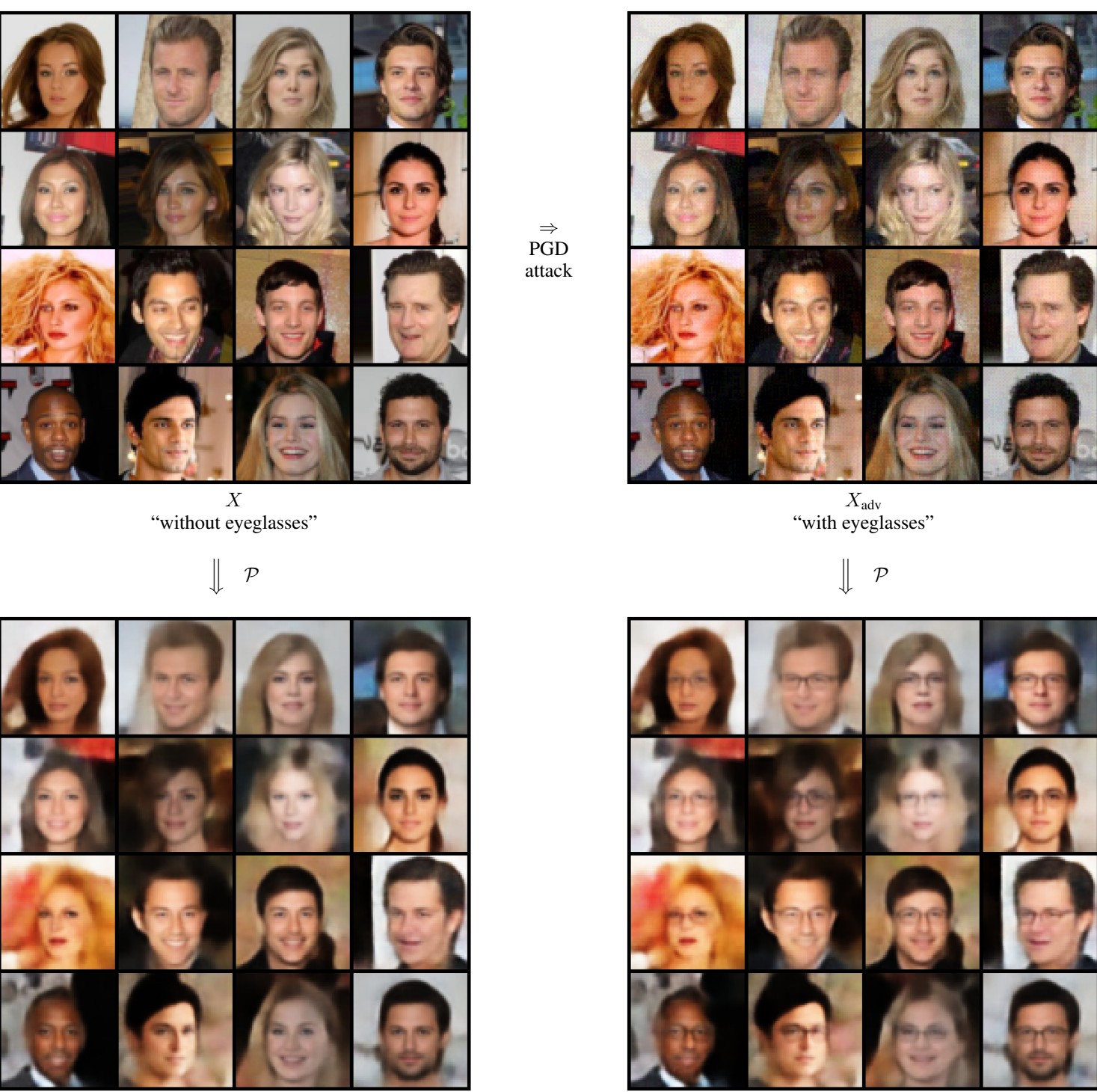

Figure 12: Visualization of adversarial examples on CelebA dataset with Eyeglasses labels. The adversarial examples are obtained by applying 100 iterations of PGD attack with $\ell_\infty$ perturbation and $\epsilon = 0.03$. The regularization parameter is set to $\lambda = 10^{-3}$.

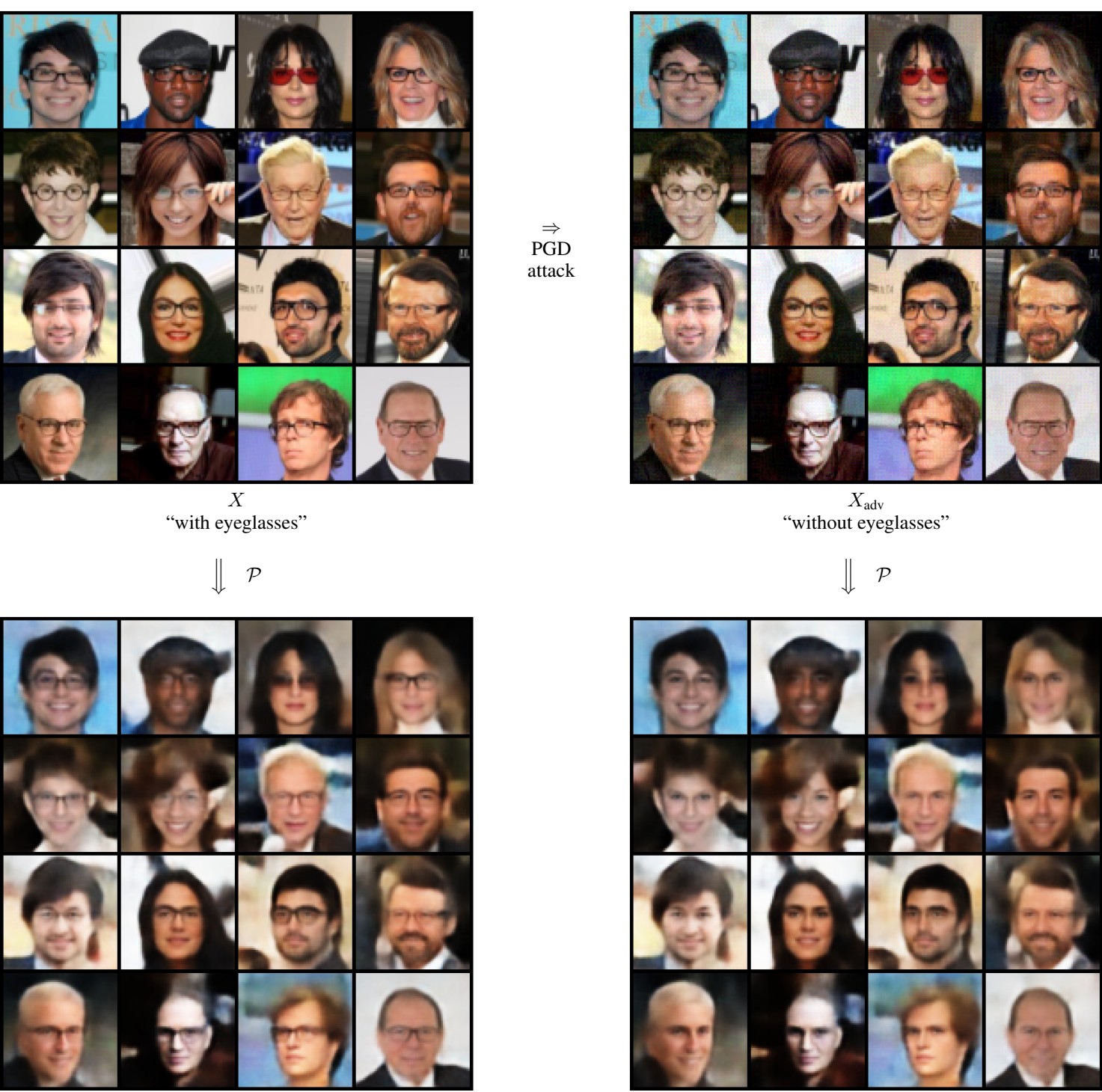

⇒
PGD
attack

$X$
"with eyeglasses"

$X_{\mathrm{adv}}$
"without eyeglasses"

$\Downarrow \mathcal{P}$

$\Downarrow \mathcal{P}$

Figure 13: Visualization of adversarial examples on CelebA dataset with Eyeglasses labels. The adversarial examples are obtained by applying 100 iterations of PGD attack with $\ell_\infty$ perturbation and $\epsilon = 0.03$. The regularization parameter is set to $\lambda = 10^{-3}$.

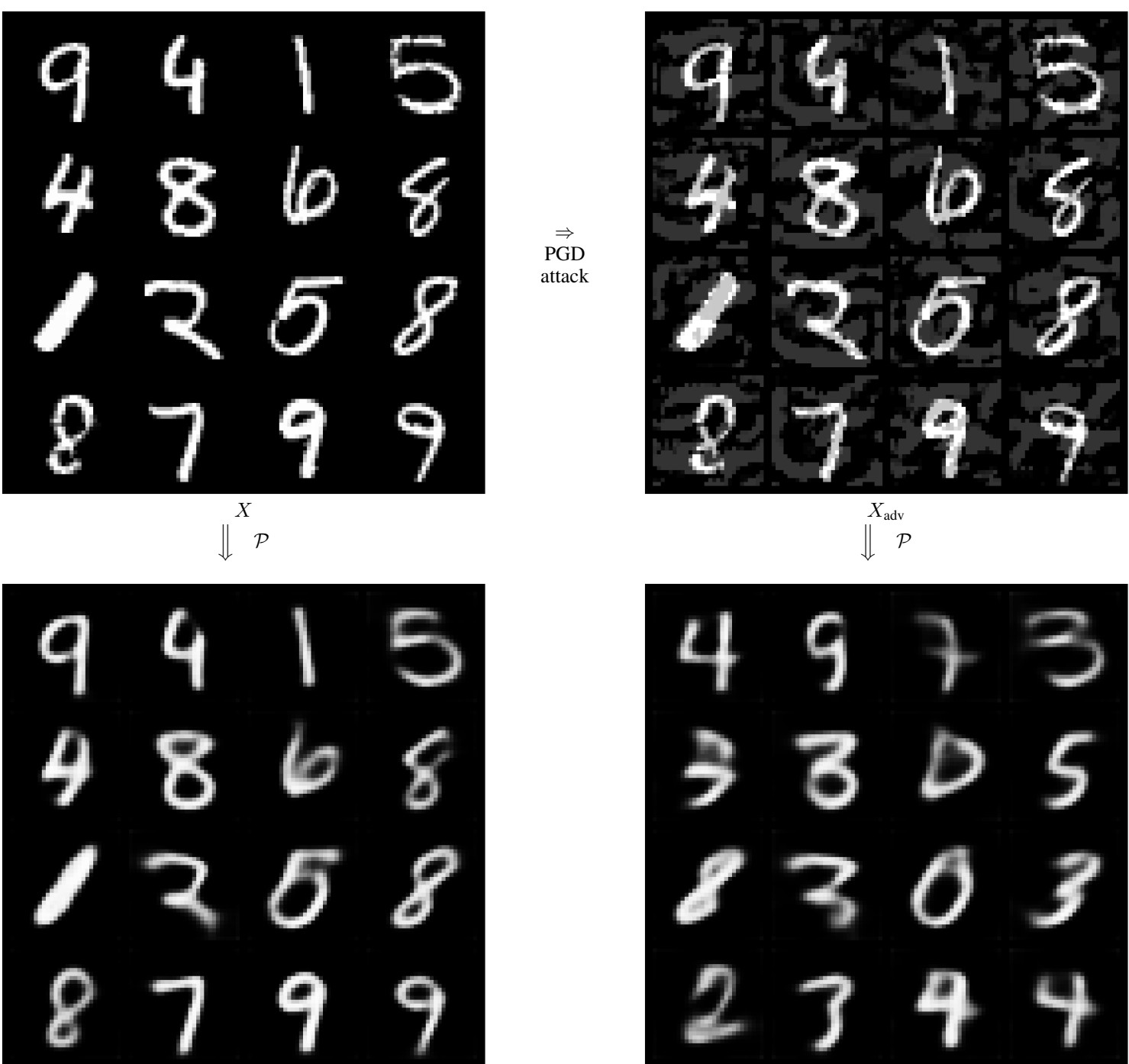

Figure 14: Visualization of adversarial examples on MNIST dataset with $\epsilon = 0.2$ ($\ell_\infty$ perturbation). The regularization parameter is set to $\lambda = 10^{-2}$.

### A.1 Some failure cases

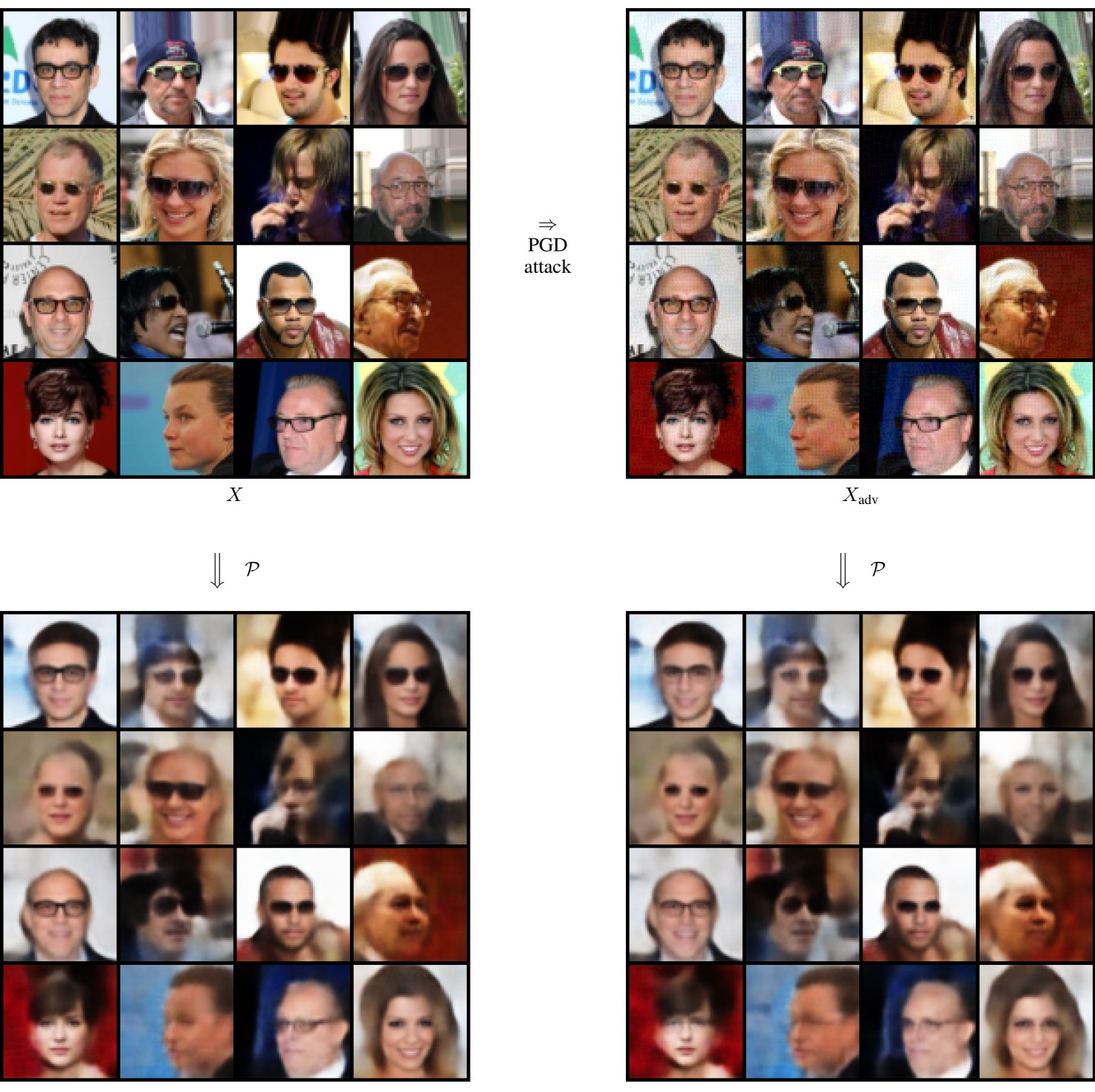

Figure 15: Visualization of adversarial examples on CelebA dataset with Eyeglasses labels. The adversarial examples are obtained by applying 100 iterations of PGD attack with $\ell_\infty$ perturbation and $\epsilon = 0.03$. The regularization parameter is set to $\lambda = 10^{-3}$.

# B  ARCHITECTURE DETAILS

## B.1  EXPERIMENTAL SETTING ON CELEBA

For the CelebA experiments, we resize the images to $64 \times 64$ and we subsample the dataset to obtain a relatively balanced dataset on the labels we are considering : Eyeglasses/Smile/Gender. The statistics of the training set and testing set are as follows:

|  | Size | Eyeglasses | Smile | Gender |
|---|---|---|---|---|
| Training Set | 20000 | 50% | 44.72% | 59.60% |
| Test Set | 6386 | 50% | 43.83% | 59.51% |

Figure 16: The percentage indicates the proportion of positive labels among the dataset.

We use Adam optimizer (Kingma & Ba, 2015) with $\beta = (0.9, 0.999)$, learning rate $10^{-3}$ in the first 20 iterations and $10^{-4}$ for later iterations. No data augmentation is used. The architecture used in the perception regularization is similar to the one from DCGAN (Radford et al., 2016). The detailed architecture is as follows:

| Input: RGB image $x \in \mathbb{R}^{3 \times 64 \times 64}$ |
|---|
| $4 \times 4$ conv. $128ch$, ReLU, stride 2, padding 1 |
| $4 \times 4$ conv. $256ch$, ReLU, stride 2, padding 1 |
| $4 \times 4$ conv. $512ch$, ReLU, stride 2, padding 1 |
| $4 \times 4$ conv. $1024ch$, ReLU, stride 2, padding 1 |
| $4 \times 4$ conv. $128ch$, ReLU, stride 1, padding 0 |
| Output: latent vector $z \in \mathbb{R}^{128}$ |

(a) Feature Map $\mathcal{F}$

| Input: latent vector $z \in \mathbb{R}^{128}$ |
|---|
| $4 \times 4$ deconv. $128ch \to 1024ch$, ReLU, stride 1, padding 0 |
| $4 \times 4$ deconv. $1024ch \to 512ch$, ReLU, stride 2, padding 1 |
| $4 \times 4$ deconv. $512ch \to 256ch$, ReLU, stride 2, padding 1 |
| $4 \times 4$ deconv. $256ch \to 128ch$, ReLU, stride 2, padding 1 |
| $4 \times 4$ deconv. $128ch \to 3ch$, ReLU, stride 2, padding 1 |
| Sigmoid |
| Output: RGB image reconstruction $x' \in \mathbb{R}^{3 \times 64 \times 64}$ |

(b) Decode $\mathcal{D}$

| Input: latent vector $z \in \mathbb{R}^{128}$ |
|---|
| Fully connected $128 \to 256$, ReLU, |
| Linear $256 \to 2$ |
| Output: prediction $y \in \mathbb{R}^{2}$. |

(c) Classifier $\mathcal{C}$

## B.2 Experimental setting on MNIST

For training the perceptual regularization, we perform 20 iterations of Adam optimizer (Kingma & Ba, 2015) with $\beta = (0.9, 0.999)$ and learning rate $10^{-3}$. No data augmentation is used.

| Input: Greyscale image $x \in \mathbb{R}^{1 \times 28 \times 28}$ |
| :---: |
| $4 \times 4$ conv. $16ch$, ReLU, stride 2, padding 1 |
| $4 \times 4$ conv. $16ch$, ReLU, stride 2, padding 1 |
| $4 \times 4$ conv. $32ch$, ReLU, stride 1, padding 0 |
| $4 \times 4$ conv. $32ch$, ReLU, stride 1, padding 0 |
| Output: latent vector $z \in \mathbb{R}^{32}$ |

(d) Feature Map $\mathcal{F}$

| Input: latent vector $z \in \mathbb{R}^{32}$ |
| :---: |
| $4 \times 4$ deconv. $32ch \to 32ch$, ReLU, stride 1, padding 0 |
| $4 \times 4$ deconv. $32ch \to 32ch$, ReLU, stride 1, padding 0 |
| $4 \times 4$ deconv. $32ch \to 16ch$, ReLU, stride 2, padding 1 |
| $4 \times 4$ deconv. $16ch \to 1ch$, ReLU, stride 2, padding 1 |
| Sigmoid |
| Output: Greyscale image reconstruction $x' \in \mathbb{R}^{1 \times 28 \times 28}$ |

(e) Decode $\mathcal{D}$

| Input: latent vector $z \in \mathbb{R}^{32}$ |
| :---: |
| Fully connected $32 \to 64$, ReLU, |
| Linear $64 \to 10$ |
| Output: prediction $y \in \mathbb{R}^{10}$. |

(f) Classifier $\mathcal{C}$

## B.3 Experimental setting on SVHN

For the SVHN experiments, the inputs are RGB images of size $32 \times 32$. No data augmentation is used. Our architecture for the perception regularization is as follows:

| Input: RGB image $x \in \mathbb{R}^{3 \times 32 \times 32}$ |
| :---: |
| $4 \times 4$ conv. $16ch$, ReLU, stride 2, padding 1 |
| $4 \times 4$ conv. $32ch$, ReLU, stride 2, padding 1 |
| $4 \times 4$ conv. $64ch$, ReLU, stride 2, padding 1 |
| $4 \times 4$ conv. $64ch$, ReLU, stride 1, padding 0 |
| Output: latent vector $z \in \mathbb{R}^{64}$ |

(g) Feature Map $\mathcal{F}$

| Input: latent vector $z \in \mathbb{R}^{64}$ |
| :---: |
| $4 \times 4$ deconv. $128ch \rightarrow 64$, ReLU, stride 1, padding 0 |
| $4 \times 4$ deconv. $64ch \rightarrow 32$, ReLU, stride 2, padding 1 |
| $4 \times 4$ deconv. $32ch \rightarrow 16$, ReLU, stride 2, padding 1 |
| $4 \times 4$ deconv. $16ch \rightarrow 3ch$, ReLU, stride 2, padding 1 |
| Sigmoid |
| Output: RGB image reconstruction $x' \in \mathbb{R}^{3 \times 32 \times 32}$ |

(h) Decode $\mathcal{D}$

| Input: latent vector $z \in \mathbb{R}^{64}$ |
| :---: |
| Fully connected $64 \rightarrow 128$, ReLU, |
| Linear $128 \rightarrow 10$ |
| Output: prediction $y \in \mathbb{R}^{10}$. |

(i) Classifier $\mathcal{C}$

