# OpenReview forum: "Perceptual Regularization: Visualizing and Learning Generalizable Representations"
_ICLR.cc/2020/Conference — Reject_

### Official Review · AnonReviewer2 · 2019-10-13
**Official Blind Review #2**

**Rating:** 6

**Review:**

Claims: The authors present perceptual regularization as a method for learning a visualization of deep representations for promoting interpretability and understanding of vulnerability to adversarial attacks. Second, they show their method can explain negative transfer to new tasks. Finally, they show that the representations learned with this regularization method transfer to unseen tasks better than without the reconstruction regularization.

Decision: Weak accept. The authors tackle an important problem with a very simple regularization technique and show how it can aid interpretability in adversarial attacks and improve transferability through strengthening attention of features to new tasks and in the multi-task setting.  While the two problems are important, results on multiple datasets would be necessary to strengthen the paper. Section 3 on perception claims to give results on MNIST and CelebA datasets, but there only seem to be results on CelebA. For transfer in section 5, MNIST or SVHN should also be included by removing classes of digits and checking performance. The conclusion addresses the use of different types of generative models, but I would be extremely interested to see how VAE compares with a deterministic auto encoder, as they are known to have much better generative properties.

For Fig 6, it would also be nice to know how robust these numbers are to lambda, and include some variation for different values of lambda, or at least some discussion of how much tuning is required.

**Experience Assessment:**

I have read many papers in this area.

**Review Assessment: Checking Correctness Of Derivations And Theory:**

N/A

**Review Assessment: Checking Correctness Of Experiments:**

I assessed the sensibility of the experiments.

**Review Assessment: Thoroughness In Paper Reading:**

I read the paper at least twice and used my best judgement in assessing the paper.

---

> ### Author Response · Authors · 2019-11-11
> **Re: Review #2**
>
> We appreciate your thorough review and constructive comments. Thank you very much for your kind support. The following are our responses to your concerns:
>
> Q: Section 3 on perception claims to give results on MNIST and CelebA datasets, but there only seem to be results on CelebA.
>
> The result on MNIST are included in the appendix due to page constraint. We will make it clear in the update version.
>
> Q: The conclusion addresses the use of different types of generative models, but I would be extremely interested to see how VAE compares with a deterministic auto encoder, as they are known to have much better generative properties.
>
> Thank you for the suggestion. We are actively following up on this extension using more powerful generators such as GANs. We expect the reconstructions to improve on auto-encoders, but  we believe that the qualitative observations and the main conceptual components are already in place in the auto-encoder setting.

---

### Official Review · AnonReviewer1 · 2019-10-14
**Official Blind Review #1**

**Rating:** 1

**Review:**

Summary:
In this study, the authors propose a new architecture for visualizing the latent space of a network. This architecture is achieved by appending a secondary decoder 'head' to visualize the latent space by reconstructing the inputs.

In summary, I found the paper to provide a simple architectural change for attempting to address this issue of network interpretability and improve transfer learning. The results all appear sensible and expected, but the experiments are somewhat weak. My primary concerns are:

1. The network architecture described is not terribly novel as many papers have explored pairing an auto-encoder with a classification task. Thus, I don't find the methods to be much of a contribution. The main contribution of the paper is thus in the interpretation and analysis of the method.

2. The results on network interpretability are all qualitative.
That is, I must make a judgement on the resulting reconstructed image. I find this unsatisfactory for providing any quantitative assessment of the performance of the method. In particular, I would want to see a quantitative assessment of this method and compare it with other network introspection techniques. Currently, there is no benchmark nor any other methods to compare against.

3. The results on transfer learning are quantitative but weak.
It is not clear to me how other simple transfer learning methods may perform on the task presented. That is, I do not know how hard the actual task is. (Would retraining a logistic classifier  with SIFT features work just as well?) I would expect to see far more experimental studies to justify these claims. Additionally, I would expect to see comparisons with other transfer learning techniques to see how well this method fares.

Major Comments:

- Why do the authors allow for the decoder to regularize the latent space? Why not just stop the gradients at the latent space to provide a microscope to visualize the latent space without effecting it?

- This reference appears to be in the same spirit as this paper and probably should be cited.
Alain, Guillaume, and Yoshua Bengio.
"Understanding intermediate layers using linear classifier probes."
arXiv preprint arXiv:1610.01644 (2016).


- Figure 2, 3 and 4 provide very nice qualitative demonstrations of the reconstruction providing insight into how an image was misclassified, however some issues remain:
  1) A human must interpret the reconstruction. This is a subjective process and is not systematic.
  2) Can you see examples in cross-validated test images where the visualization would predict a misclassification? Can you see counter-examples where the resulting image does not make sense?

- Figure 6 and 7 provide interesting results indicating that the unsupervised objective improves transfer learning across several classification tasks related to faces. The results are encouraging but I am concerned that (a) the task is too easy, (b) tuning the hyperparameter for \lambda can be difficult. For (b), I would like to ensure that \lambda was tuned independent on a third validation test (and not the test set whose numbers are reported). For (a), I suspect that an important benchmark is to just take set \lambda = \infty and then perform linear classification on the embedding. It would be interesting to see how well these numbers fare as a baseline compared to the reported numbers.

**Experience Assessment:**

I have published one or two papers in this area.

**Review Assessment: Checking Correctness Of Derivations And Theory:**

I assessed the sensibility of the derivations and theory.

**Review Assessment: Checking Correctness Of Experiments:**

I carefully checked the experiments.

**Review Assessment: Thoroughness In Paper Reading:**

I read the paper at least twice and used my best judgement in assessing the paper.

---

> ### Author Response · Authors · 2019-11-11
> **Re: Review #1 Part 1/2**
>
> We appreciate your thorough review and constructive comments. Our response to your concerns are the following:
>
> Q: Why do the authors allow for the decoder to regularize the latent space?
>
> One reason we consider it as a regularization is to enhance the information retained by the network. The quality of the reconstruction images is a measure of how much information is kept in the representation. As shown in Figure 5 (in section about attention), colors and other shapes started to emerge as we increase the regularization parameter. This justifies that the learned representation by our method contains more information. Moreover, more informative implies better transferability, as shown in the transfer learning experiments.
>
> Q: Can you see examples in cross-validated test images where the visualization would predict a misclassification? Can you see counter-examples where the resulting image does not make sense?
>
> a) There are a small fraction of failure cases on visualizing adversarial examples where the reconstructed image of adversarial examples P(X_{adv}) are similar to the reconstruction of original images P(X) (which means no human-observable semantic difference). We will include some examples in the appendix in the updated version.
> b) The reviewer’s remark on misclassification is very relevant. Indeed, we could consider using the distance between the original image and the reconstruction image as a measure to detect outlier/adversarial examples, as shown in figure 8. We are actively following up on this idea as future work.
>
> Q: I suspect that tuning the hyper-parameter for \lambda can be difficult
>
> We perform additional experiments on performance while varying the parameter \lambda. We see that there is no essential difference in the performance for a large range of parameter choices from 10^{-5} to 10^{-2}. Results will be included in the updated version.

---

> > ### Author Response · Authors · 2019-11-11
> > **Re: Review #1 Part 2/2**
> >
> > Q:  The network architecture described is not terribly novel
> >
> > We agree with the reviewer adding an auto-encoder is not a novel idea and we have specifically mentioned in the paper that “There has been previous work on the idea of combining generative and discriminative models (Lasserre et al., 2006; Grabner et al., 2007; Larochelle & Bengio, 2008; Le et al., 2018)”. The main contribution of our paper is to apply it to obtain interesting observations which are completely novel:
> > a) Visualization of the effect of adversarial attack on the latent encoding, giving visual confirmation that the attack perturbs the encoding into a region corresponding to the attack’s target label.
> > b) We provide a brand new method for latent space interpolation. Many entire papers are written on this topic, such as [1,2], and involve significant manual effort to identify which latent direction corresponds to a given semantic feature. Our method, on the other hand, gives an automatic way of achieving this using adversarial attacks. Our simple and effective method is an important contribution not acknowledged in the reviews.
> > c) The illustration on an intrinsic problem in supervised learning paradigm: the representation learned is biased by the supervised learning task. Moreover, our section 4 (on attention) provides another visual explanation, showing that multi-class supervised learning forgets many important details of the image - such as color and certain shapes - that are not relevant to the learning task. This is supported by section 4.5 of [3] which shows that ImageNet features may not transfer well to fine grained tasks.
> >
> > Finally, there are a large amount of stylistic improvements we will make, including polishing the writing and adding references. However, none of these issues are integral to the message and contribution of our paper and we will make these changes for an updated version.
> >
> >
> > [1] Jahanian, A., Chai, L. and Isola, P., 2019. On the''steerability" of generative adversarial networks. arXiv preprint arXiv:1907.07171.
> >
> > [2] Radford, A., Metz, L. and Chintala, S., 2016. Unsupervised representation learning with deep convolutional generative adversarial networks. ICLR
> >
> > [3] Kornblith, S., Shlens, J. and Le, Q.V., 2019. Do better imagenet models transfer better?. CVPR

---

### Official Review · AnonReviewer4 · 2019-10-30
**Official Blind Review #4**

**Rating:** 3

**Review:**

Summary: This paper discusses the use of a reconstruction network which is fed an internal representation of a deep network and is trained to reconstruct the input. The classification and reconstruction networks are trained simultaneously with a combined loss function. Various experiments are carried out to test whether this approach can help investigate adversarial examples and the quality of learned features for transfer learning.

Review: The idea presented in this paper is simple and easy to understand and the paper includes experiments testing its effectiveness in various settings. However, there are various issues with the paper in its current form: First, the idea of reconstructing hidden activations is not new and the paper fails to cite future work. "Dynamic Routing Between Capsules" by Sabour et al. consider adding a similar reconstruction network to Capsule networks. "Detecting and Diagnosing Adversarial Images with Class-Conditional Capsule Reconstructions" considers the use of this reconstruction network for detecting adverarsarial examples (like this paper) and also includes an unconditional reconstruction network which is added to a convnet (like this paper). I imagine there are other examples, but those are ones I know of. Second, the experiments are limited - they only include simple datasets (SVHN, MNIST, CelebA) which are normalized so that the object appears in the center of the image. This makes reconstructions difficult. The experiments also lack various important details like the model architectures and other hyperparameters, and overall are mostly "proof of concept" experiments rather than exhaustively testing out the proposed method. Finally, some of the claims made in the paper (particularly in the introduction) are dubious, for example that adversarial examples are a security concern and that transfer learning has been unsuccessful because it can harm performance. Based on these issues, I won't recommend acceptance. I think if the authors better situated the novelty of their work, included more exhaustive and difficult experiments, and/or focused on one specific application area it would be a stronger submission.

Specific comments:
- "to be understandable" - can you be more specific? Internal representations of deep neural networks are not understandable whatsoever on their own, we usually use some post-hoc method (like your reconstruction network) to try to interpret them somehow.
- "[adversarial examples] raise many security concerns, such as the reliability of driverless cars, or the trustworthiness of facial recognition systems" The field is moving towards a consensus that adversarial examples do not pose a realistic security threat in any of the settings they have been tested in. The most significant description of this viewpoint is in "Motivating the Rules of the Game for Adversarial Example Research" by Gilmer et al.
- "but has made the transferability of features to new tasks difficult, sometimes even harming performance" This statement completely omits a huge body of work showing how effective transfer learning is, not only in computer vision (typically going from an ImageNet classifier to some downstream task with a small training set) but also in NLP (where transfer learning-based methods have state-of-the-art in virtually every benchmark). A good discussion of when transfer learning works and doesn't in CV is provided in "Do Better ImageNet Models Transfer Better?" by Kornblith et al.
- The variables X and Y (without mathcal) in the "(Perceptual regularization)" definition equation are not defined.
- The classification and reconstruction models are not described in Section 3. This is incredibly important. The quality of the reconstructions is likely extremely dependent on these architectural choices as they provide an implicit prior over the image space. This detail needs to be filled in. E.g. the pooling/downsampling in your classification network  will have a strong impact on the reconstruction quality, as will the size, depth, and configuration of the reconstruction network. Furthermore, I imagine the regularization parameter lambda is extremely important (as shown in section 4.2) but its value is not mentioned.
- The experiments are on SVHN, MNIST, and CelebA which are all "normalized" datasets, i.e. all of their constituent images are normalized so that the object of interest appears in the same position in the center of the image. This makes reconstruction much, much easier. Results should be presented on non-normalized datasets (e.g. CIFAR-10, ImageNet, etc) in order to be convincing.
- "But, as the network gets more complicated it becomes computationally difficult to decode the feature map." Can you quantify this claim or back it up with experiments?
- While the results on transferring to different CelebA classification tasks are interesting, it is much more common to pre-train on a diverse task like ImageNet than an extremely fine-grained task like classifying smiling or not smiling. I think these results would be more impressive if you followed modern experimental practice more closely.
- "The cost is that our method performs slightly poorer than the standard training method when S = T" I would not call a difference of upwards of 8% absolute small.

**Experience Assessment:**

I have published one or two papers in this area.

**Review Assessment: Checking Correctness Of Derivations And Theory:**

N/A

**Review Assessment: Checking Correctness Of Experiments:**

I assessed the sensibility of the experiments.

**Review Assessment: Thoroughness In Paper Reading:**

I read the paper thoroughly.

---

> ### Author Response · Authors · 2019-11-11
> **Re: Review #4**
>
> We appreciate your thorough review and constructive comments. Our response to your concerns are following:
>
> Q:  I think if the authors better situated the novelty of their work, included more exhaustive and difficult experiments, and/or focused on one specific application area it would be a stronger submission.
>
> A: The main contribution of our paper is to apply a simple method to obtain interesting observations which are completely novel:
> a) Visualization of the effect of adversarial attack on the latent encoding, giving visual confirmation that the attack perturbs the encoding into a region corresponding to the attack’s target label. This sheds new light and helps build understanding on this active area of research.
> b) We provide a brand new method for latent space interpolation. Many entire papers are written on this topic, such as [1,2], and involve significant manual effort to identify which latent direction corresponds to a given semantic feature. Our method, on the other hand, gives an automatic way of achieving this using adversarial attacks. Our simple and effective method is an important contribution not acknowledged in the reviews.
>
> Q: The experiments are limited - they only include simple datasets (SVHN, MNIST, CelebA) which are normalized so that the object appears in the center of the image. This makes reconstructions difficult.
>
> A: CelebA, SVHN, MNIST are standard datasets on which the experimental results of many papers are based, including the reference suggested by the reviewer [3]. We believe that, despite the existence of more difficult datasets, our results stand on their own. We have preliminarily tried our method on CIFAR-10 and the reconstruction quality does degrade a lot. We believe this is due to the capacity of the generator, since it is commonly known that auto-encoders do not yield good reconstructions on datasets like CIFAR-10/ImageNet. We therefore expect that a more powerful generative model, such as a GAN, will be necessary. We are actively following up on this extension as future work, but the main conceptual components is already in place in the auto-encoder setting.
>
> Q: The experiments also lack various important details like the model architectures and other hyperparameters
> A: Architecture and training details are given in full in the appendix to save space. We will include a signposts in the main body of the paper to make this clear.
>
> Q: “But, as the network gets more complicated it becomes computationally difficult to decode the feature map” Can you quantify this claim or back it up with experiments?
> A: We have experiments to justify this claim and will include them in the appendix in an updated version.
>
> Q: Some of the claims made in the paper (particularly in the introduction) are dubious
> A: There are a large amount of stylistic improvements we can, and will, make. However, none of these issues are integral to the message and contribution of our paper and we will make these changes for an updated version.
>
>
> [1] Jahanian, A., Chai, L. and Isola, P., 2019. On the''steerability" of generative adversarial networks. arXiv preprint arXiv:1907.07171.
>
> [2] Radford, A., Metz, L. and Chintala, S., 2016. Unsupervised representation learning with deep convolutional generative adversarial networks. ICLR
>
> [3] Qin, Y.,  Frosst, N.,  Sabour, S., Raffel, C., Cottrell G., and Hinton G.  2019 Detecting and Diagnosing Adversarial Images with Class-Conditional Capsule Reconstructions

---

> > ### Comment · AnonReviewer4 · 2019-11-15
> > **Response**
> >
> > Thank you for your clarifications. Regarding the novelty of the paper (which is the main criticism),
> >
> > > The main contribution of our paper is to apply a simple method to obtain interesting observations which are completely novel: a) Visualization of the effect of adversarial attack on the latent encoding
> >
> > This is not a novel insight, similar findings are available in Qin et al.
> >
> > >  Since it is commonly known that auto-encoders do not yield good reconstructions on datasets like CIFAR-10/ImageNet
> >
> > This is not true, there have been plenty of autoencoder models which obtain good results on datasets as complicated as ImageNet, e.g. "Adversarial Feature Learning" by Donahue et al., "PixelVAE: A Latent Variable Model for Natural Images" by Gulrajani et al., "Improved Variational Inferencewith Inverse Autoregressive Flow" by Kingma et al., etc. I think what you actually mean is that "auto-encoders *trained with a mean-squared-error loss* do not yield good reconstructions..." which is somewhat true, but that's exactly my point.
> >
> > I remain confident in my rating. I think if the paper was rewritten to focus on a specific (hopefully novel) application of this basic idea, it could be greatly improved.

---

### Author Response · Authors · 2019-11-15
**Modifications based on the reviews**

Dear Reviewers,

Based on your thoughtful recommendations we have decided to change the focus of the paper to concentrate on visualizing internal representations, with applications to adversarial attacks, model diagnostics, and latent space interpolation. In particular we have expanded the section on latent space interpolation. We have removed the section on transfer learning and leave it to future work to more thoroughly experiment with that idea. We have also included additional references and context on related work so as to situate our contribution more clearly.

---

### Decision · Program_Chairs · 2019-12-19

**Decision:**

Reject

**Comment:**

This paper proposes a new mechanism to visualize the latent space of a neural network. The idea is simple and the paper includes several experiments to test the effectiveness of the method. However, the method bears similarity to previous work and the evaluation does not sufficiently show quantitative improvements over other introspection techniques. The reviewers found this was a substantial problem and for this reason the paper is not ready for publication. The paper should improve its discussion of prior work and better establish its place in this regard.